# The expansion in lymphoid organs of IL-4+ BATF+ T follicular helper cells is linked to IgG4 class switching in vivo

Takashi Maehara[1],*, Hamid Mattoo[1],*, Vinay S Mahajan[1] , Samuel JH Murphy[1] , Grace J Yuen[1], Noriko Ishiguro[2], Miho Ohta[2], Masafumi Moriyama[2], Takako Saeki[3] , Hidetaka Yamamoto[4,5], Masaki Yamauchi[2], Joe Daccache[1], Tamotsu Kiyoshima[6], Seiji Nakamura[2], John H Stone[7], Shiv Pillai[1]

Distinct T follicular helper (T$_{FH}$) subsets that influence specific class-switching events are assumed to exist, but the accumulation of isotype-specific T$_{FH}$ subsets in secondary lymphoid organs (SLOs) and tertiary lymphoid organs has not been hitherto demonstrated. IL-4–expressing T$_{FH}$ cells are surprisingly sparse in human SLOs. In contrast, in IgG4-related disease (IgG4-RD), a disorder characterized by polarized Ig class switching, most T$_{FH}$ cells in tertiary and SLOs make IL-4. Human IL-4+ T$_{FH}$ cells do not express GATA-3 but express nuclear BATF, and the transcriptomes of IL-4–secreting T$_{FH}$ cells differ from both PD1$^{hi}$ T$_{FH}$ cells that do not secrete IL-4 and IL-4–secreting non-T$_{FH}$ cells. Unlike IgG4-RD, IL-4+ T$_{FH}$ cells are rarely found in tertiary lymphoid organs in Sjögren's syndrome, a disorder in which IgG4 is not elevated. The proportion of CD4+IL-4+BATF+ T cells and CD4+IL-4+CXCR5+ T cells in IgG4-RD tissues correlates tightly with tissue IgG4 plasma cell numbers and plasma IgG4 levels in patients but not with the total plasma levels of other isotypes. These data describe a disease-related T$_{FH}$ subpopulation in human tertiary lymphoid organs and SLOs that is linked to IgG4 class switching.

## Introduction

T follicular helper (T$_{FH}$) cells provide help to B cells during T-dependent immune responses, and they contribute to isotype switching, somatic hypermutation, germinal center (GC) formation, and the selection of high-affinity B cells in the GC (Vinuesa et al, 2005; King et al, 2008; Crotty, 2011). However, how exactly T$_{FH}$ cells provide specificity to class-switching events remains unclear. The idea that unique T$_{FH}$ subsets separately and specifically drive class switching to different Ig isotypes is attractive, but no in vitro or in vivo data exist to firmly establish this notion. Indeed, there have been no studies using multicolor staining approaches to examine human T$_{FH}$ cells in situ in secondary lymphoid organs (SLOs) or tertiary lymphoid organs (TLOs). The possibility that chronic disease states exhibiting polarized isotype switching could provide novel insights into specialized T$_{FH}$ cells served as the rationale for undertaking this study.

Some evidence for specialized T$_{FH}$ subsets, albeit indirect, comes from the studies of circulating human T$_{FH}$ cells that have described three T$_{FH}$ subsets defined on the basis of chemokine receptor expression patterns. The relationship between blood T$_{FH}$-cell subsets and T$_{FH}$ cells in SLOs or TLOs remains unclear. In the studies of Ueno et al (Morita et al, 2011; Ueno et al, 2015) on blood T$_{FH}$ subsets, T$_{FH1}$ cells secrete IFN-γ upon activation and have limited isotype-switching activity when examined in in vitro coculture experiments. T$_{FH2}$ cells secrete IL-4 after many days of in vitro stimulation and can mediate class switching to IgA, IgE, and essentially all IgG isotypes, including IgG4. T$_{FH17}$ cells secrete IL-17 following activation and are equally promiscuous.

Although all T$_{FH}$ cells may have the potential to secrete IL-4, one report has described polarized IL-4–secreting T$_{FH}$ cells in mice in the context of an allergic disease model, and it was suggested that these cells could subsequently differentiate into T$_{H2}$ cells (Ballesteros-Tato et al, 2016). An illuminating study using reporter mice has led to the view that T$_{FH}$ cells initially make IL-21, mature into cells that make IL-21 and IL-4, and eventually make IL-4 alone (Weinstein et al, 2016). These studies also demonstrated that the use of a type 2 response–linked murine pathogen facilitated the induction of IL-4–secreting "T$_{FH4}$" cells. There have been no other reports establishing the existence of functionally distinct T$_{FH}$ subsets in human or murine SLOs or TLOs. Moreover, no cytokine-expressing subset of these cells in tissue sites has been linked so far to any specific disease,

[1]Ragon Institute of MGH, MIT, and Harvard, Massachusetts General Hospital, Harvard Medical School, Boston, MA, USA  [2]Section of Oral and Maxillofacial Oncology, Division of Maxillofacial Diagnostic and Surgical Sciences, Faculty of Dental Science, Kyushu University, Fukuoka, Japan  [3]Department of Internal Medicine, Nagaoka Red Cross Hospital, Nagaoka, Japan  [4]Division of Diagnostic Pathology, Kyushu University Hospital, Fukuoka, Japan  [5]Department of Anatomic Pathology, Kyushu University, Fukuoka, Japan  [6]Laboratory of Oral Pathology, Division of Maxillofacial Diagnostic and Surgical Sciences, Faculty of Dental Science, Kyushu University, Fukuoka, Japan  [7]Division of Rheumatology, Allergy, and Immunology, Massachusetts General Hospital, Harvard Medical School, Boston, MA, USA

Correspondence: pillai@helix.mgh.harvard.edu
*Takashi Maehara and Hamid Mattoo contributed equally to this work.

nor have $T_{FH}$ subsets been defined that determine specific polarized class-switching events. How the overall transcriptome of an IL-4–secreting $T_{FH}$-cell population may differ from other $T_{FH}$ cell types has also never been determined because such cells have not previously been purified from SLOs or TLOs.

IgG4-related disease (IgG4-RD) is a chronic inflammatory disease characterized by tumescent lesions with characteristic storiform fibrosis, obliterative phlebitis, and a marked lymphoplasmacytic infiltrate that includes a large proportion of IgG4-positive plasma cells (Mahajan et al, 2014; Kamisawa et al, 2015). Circulating expansions of plasmablasts, most of which express IgG4, are a hallmark of active disease (Mattoo et al, 2014). We have shown that circulating plasmablasts are heavily somatically hypermutated, implying that these B-lineage cells are derived from GCs. We have also shown that patients with IgG4-RD exhibit large clonal expansions of CD4+ CTLs, the dominant T cells in disease tissues, and that these cells are activated at lesional sites, where they secrete IL-1β, TGF-β1, and IFN-γ (Mattoo et al, 2016; Maehara et al, 2017). Although an increase in blood $T_{FH2}$ cells has been noted in IgG4-RD (Akiyama et al., 2015, 2016), these cells function promiscuously in vitro, as they can facilitate class switching to multiple isotypes in coculture experiments. There is no evidence so far connecting blood $T_{FH}$-cell subsets to any functional counterparts in SLOs or TLOs.

Multicolor staining approaches have hitherto not been used to examine lymphoid organ $T_{FH}$ cells in situ. We show here, using such an approach, that $T_{FH}$ cells making IL-4 are surprisingly sparse in normal human tonsils and mesenteric lymph nodes. However, IL-4– and BATF-expressing $T_{FH}$ cells are dramatically expanded in IgG4-RD TLOs and lymph nodes, frequently making up more than half of all $T_{FH}$ cells. This subset of human $T_{FH}$ cells can be found in association with activation-induced cytidine deaminase (AID)-expressing B cells, is more frequent in extrafollicular sites than in the light zone, and is linked to specific class switching to IgG4.

Isolating T-cell subsets based on their function, such as secretion of their cardinal cytokine, and characterizing their gene expression profiles can help us gain better insights into the biology of specific $T_{FH}$ subsets, as well as define better surface markers for their flow cytometric identification. We therefore used an IL-4 capture assay, followed by FACS sorting, to isolate IL-4–secreting $T_{FH}$ cells from a human tonsil and compared their transcriptomic profiles with CXCR5$^{hi}$PD1$^{hi}$IL-4$^-$ tonsillar $T_{FH}$ cells and IL-4–producing CXCR5$^-$ non-$T_{FH}$ cells ($T_{H2}$ cells). Our studies validate the notion of functionally distinct $T_{FH}$ subsets, establish a link between tissue and lymphoid organ human IL-4–secreting $T_{FH}$ cells and IgG4-RD, and identify genes that are specifically expressed in and define the human IL-4–secreting $T_{FH}$-cell subset.

# Results

### CD4$^+$ICOS$^+$IL-4$^+$ T cells are sparse in normal tonsils and lymph nodes

We initially examined normal tonsils to quantitate CD4+ICOS+ T cells that express IL-4 in situ. Almost all the IL-4–expressing cells seen in tonsils were CD4+ but did not express ICOS (Fig 1A and B). These CD4+

IL-4+ICOS$^-$ T cells are presumably $T_{H2}$ cells that reside in T-cell zones in tonsils. Quantitation revealed that CD4+ICOS+IL-4+ $T_{FH}$ cells represent <0.5% of all CD4+ICOS+ $T_{FH}$ cells in normal tonsils and lymph nodes (Fig 1C). However, quantitative analyses revealed that approximately 40% of CD4+ICOS+ $T_{FH}$ cells in an IgG4-RD patient expressed IL-4 (Fig 1C). The low frequency of $T_{FH}$ cells that express IL-4 was confirmed in 12 different tonsil specimens (Fig 1D). Relatively low frequencies of IL-4+CD4+ICOS+ and IL-4+ICOS+Bcl6+ $T_{FH}$ cells were also observed in normal mesenteric lymph nodes in addition to tonsillar samples (Fig 1E).

To more directly examine GC $T_{FH}$ cells, we initially localized GCs in 12 different human tonsils using staining for Bcl-6 and then analyzed CD4+Bcl-6+IL-4+ T cells (Fig 2A). These analyses also revealed that CD4+Bcl-6+IL-4+ GC $T_{FH}$ cells represented a very small proportion (ranging from 1% to 10%) of all GC $T_{FH}$ cells.

### CD4$^+$CXCR5$^+$IL-4$^+$ $T_{FH}$ cells are abundant in IgG4-RD but rare in normal SLOs and in Sjögren's syndrome (SS)

We analyzed SLOs from healthy individuals and affected TLOs in the submandibular glands (SMGs) from IgG4-RD patients with active disease for IL-4–synthesizing $T_{FH}$ cells and focused our analyses in the vicinity of TLO-like structures containing GCs. TLOs with GCs (Ruddle, 2014) were identified using multicolor immunofluorescence approaches (CD4, CD19, Bcl6, and DAPI). In this study, 17 of 25 patients (68%) with IgG4-RD and 7 of 15 patients (47%) with severe SS had TLOs in affected salivary glands (Table S3). GCs within TLOs were both more frequent and larger in the salivary glands from IgG4-RD than severe SS (Table S3). As shown in Fig 2B, CD4+CXCR5+IL-4+ cells were found in human tonsils and lymph nodes but were sparse. In contrast, these CD4+CXCR5+IL-4+ $T_{FH}$ cells were abundant in IgG4-RD. Quantitation of CD4+CXCR5+IL-4+ $T_{FH}$ cells revealed that approximately 67% of CD4+CXCR5+ $T_{FH}$ cells in an IgG4-RD patient expressed IL-4, whereas fewer than 5% of tonsillar CD4+CXCR5+ $T_{FH}$ cells expressed IL-4 (Fig 2B). CD4+CXCR5+IL-4+ $T_{FH}$ and CD4+ICOS+IL-4+ $T_{FH}$ cells are clearly far more abundant in IgG4-RD than in normal SLOs in or around TLOs from affected tissues from patients with SS (Fig 2B).

We extended our studies to lymph nodes from patients with IgG4-RD. Quantification revealed that almost all IL-4–expressing CD4+ T cells in IgG4-RD lymph nodes were CXCR5+ $T_{FH}$ cells (Fig 2C).

### IL-4$^+$BATF$^+$ $T_{FH}$ cells are rare in normal SLOs but abundant in IgG4-RD

The regulation of IL-4 expression in murine $T_{FH}$ cells is distinct from the mechanisms that drive IL-4 expression in $T_{H2}$ cells. BATF is a transcription factor that may regulate IL-4 secretion by murine $T_{FH}$ cells (Sahoo et al, 2015) but may more broadly help identify $T_{FH}$ cells, some of which do not necessarily express IL-4 (Weinstein et al, 2016). Many CD4+IL-4+ T cells in the vicinity of GCs in affected IgG4-RD tissues express nuclear BATF (Fig 2D). Not surprisingly, quantification revealed that these IL-4–expressing CD4+BATF+ T cells are abundant in IgG4-RD tissue lesions but rare in normal SLOs (Fig 2E). Using parallel sections of tissues from an IgG4-RD patient, we noted that CD4+BATF+IL-4+ T cells were located in the same region in which

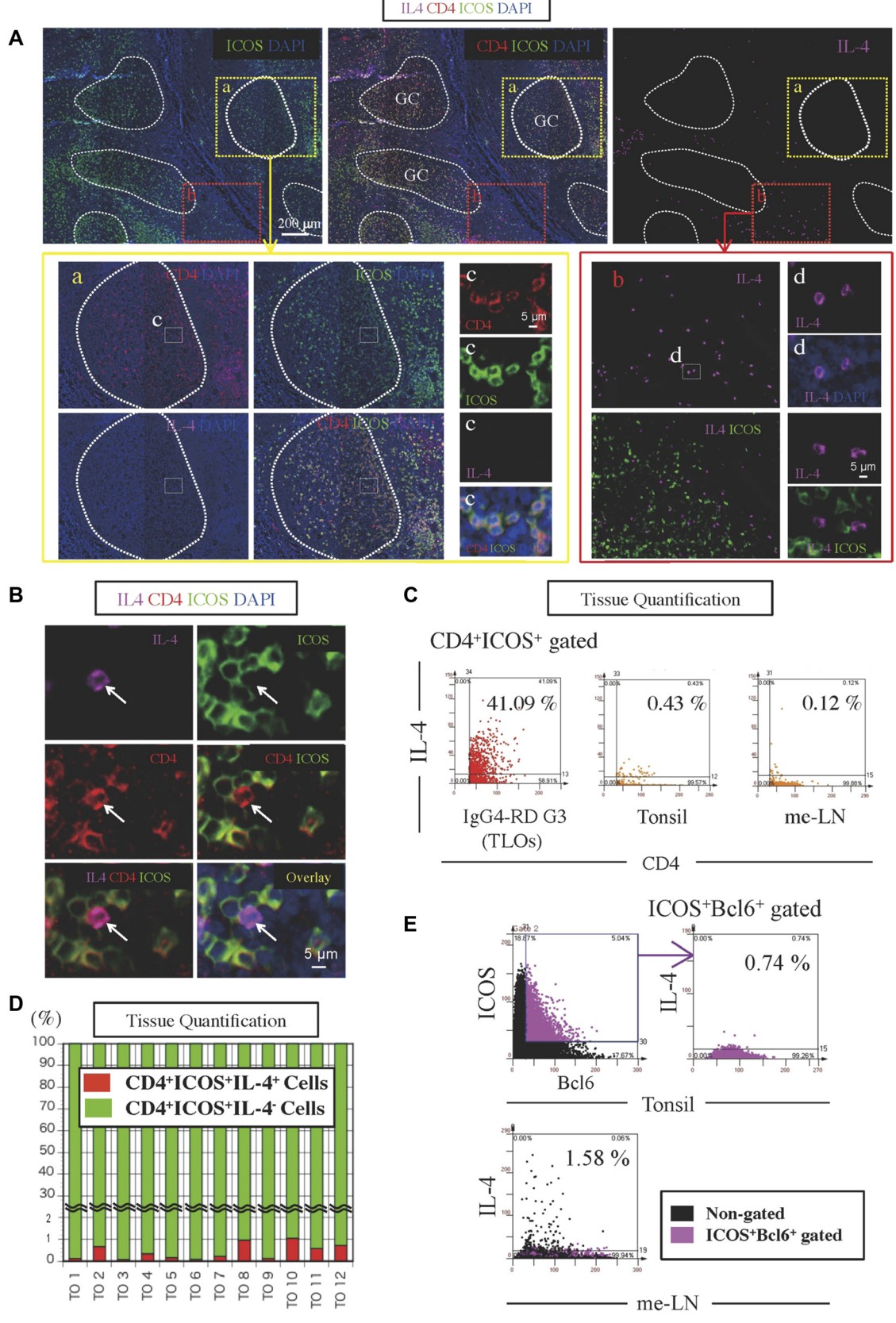

IL-4–expressing CD4$^+$CXCR5$^+$T$_{FH}$ cells were observed. Quantification of CXCR5$^+$BATF$^+$IL-4$^+$ cells revealed that approximately 60% of CXCR5$^+$IL-4$^+$ cells in an IgG4-RD patient with TLOs expressed BATF (Fig 2F).

We considered the possibility that the expansion of these IL-4–expressing T$_{FH}$ cells might represent an important disease-related T$_{FH}$ subset and contribute to the specific class-switching event in IgG4-RD. ICOS$^+$BATF$^+$ T$_{FH}$ cells are far more abundant in IgG4-RD lymph nodes than in normal tonsils (Fig 2G). As shown in Fig 2H, ICOS$^+$BATF$^+$IL-4$^+$ T$_{FH}$ cells were detected in patients with IgG4-RD, and these cells were abundant. Quantification of ICOS$^+$BATF$^+$IL-4$^+$ T$_{FH}$ cells revealed that approximately 60% of ICOS$^+$IL-4$^+$ T$_{FH}$ cells in an IgG4-RD lymph node patient expressed BATF (Fig 2I). Furthermore, quantification revealed that the majority of ICOS$^+$BATF$^+$ T$_{FH}$ cells in IgG4-RD lymph nodes expressed IL-4 as well (Fig 2I). Interestingly, quantification of CD40L$^+$BATF$^+$IL-4$^+$ T cells revealed that these cells represented approximately 98% of CD40L$^+$BATF$^+$ T cells in an IgG4-RD patient (Fig 2J). These data indicate that a large number of lesional IL-4$^+$ T$_{FH}$ cells in IgG4-RD express BATF, CD40L, and ICOS.

### IL-4–secreting T$_{FH}$ cells are a distinct population of T$_{FH}$ cells

Although visualization by multicolor staining permits anatomic localization of T$_{FH}$ cells in tissues, it can provide only limited information about a few expressed proteins in any putative T$_{FH}$-cell subset, and detailed characterization of any specific cytokine-secreting T$_{FH}$ subset is currently lacking. To better understand the biology of IL-4–secreting T$_{FH}$ cells found in lymphoid organs, we performed RNA sequence analysis of viable IL-4–secreting T$_{FH}$ cells from human tonsils. Although the fraction of IL-4–producing T$_{FH}$ cells is low in human tonsils, we were able to purify this subset by starting with 600 million tonsil cells using an IL-4 cytokine capture strategy. To promote cytokine secretion, CD19-depleted lymphocytes from tonsils were stimulated overnight with plate-coated anti-CD3 and anti-CD28 antibodies. We then compared the transcriptomes of FACS-sorted IL-4–producing T$_{FH}$ cells with those of CXCR5$^{hi}$PD1$^{hi}$ T$_{FH}$ cells that did not secrete IL-4, as well as IL-4–secreting CD45RA$^-$CXCR5$^-$ non-T$_{FH}$ cells obtained from the same tonsil (Fig 3A). CD4$^+$ CD45RA$^+$ naive cells were also included as an additional control. Of the 26,002 mapped transcripts, 7,792 were differentially expressed across the four conditions (Fig 3B). The IL-4–producing and nonproducing T$_{FH}$ cells were most similar and markedly dissimilar from IL-4–secreting non-T$_{FH}$ cells or naive CD4$^+$ T cells (Fig 3C). In contrast to the IL-4–producing non-T$_{FH}$ cells that express high levels of *IL-4*, *IL-5*, and IL-13, reflecting a T$_{H2}$ signature, IL-4–producing T$_{FH}$ cells express only IL-4 but much lower levels of IL-5 and *IL-13* (Fig 3D). To examine lineage-defining genetic regulators and functionally relevant effector molecules, we focused our analysis on differentially expressed CD molecules, transcription factors, and cytokines (Fig 3E). The transcript level of genes critical to T$_{FH}$ function such as *CD40L*, *ICOS*, *CXCR5*,

*IL-21R*, and *PD1* was highest in the IL-4–secreting T$_{FH}$ cells. In addition, they also expressed high levels of *CCR4*, *CD200*, *CTLA4*, and *GITR* and low levels of *CD6*, *CD27*, *CD28*, *SELL*, *IL-7R*, and *CD74*. Although the expression levels of these cell surface markers are derived from in vitro activation during cytokine capture, the CD markers specific for cells with an IL-4–secreting T$_{FH}$ phenotype may aid in the specific flow cytometric identification of ex vivo human IL-4–secreting T$_{FH}$ cells in future studies. As expected, CD4$^+$ CD45RA$^+$ cells had the highest level of *CCR7*, *CD27*, and *CD62L*. The high expression levels of multiple chemokines, cytokines, and their receptors including *CCR2*, *CCR6*, *CXCR3*, *CXCR6*, *IL-2RA*, *IL-2RB*, *IL-10RA*, *CCL4*, *IFNG*, *IL-2*, *IL-3*, *IL-4*, *IL-5*, *IL-9*, *IL-10*, *IL-17A*, *IL-22*, and *IL-23A* within the IL-4–producing CD4$^+$CD45RA$^-$ non-T$_{FH}$ cells perhaps indicate the heterogeneity among these cells and may reflect the overlap and plasticity between T$_{H1}$, T$_{H2}$, and T$_{H17}$ subsets that are seen following TCR stimulation in the absence of polarizing cytokines (i.e., anti-CD3 and anti-CD28 alone). Interestingly, the IL-4–producing T$_{FH}$ cells express the highest levels of *BCL6* and *BATF* in this comparison but not transcription factors related to T$_{H2}$ differentiation such as *GATA3*, *STAT5A*, and *PRDM1 (BLIMP1)*, which are instead abundantly expressed in the IL-4–producing non-T$_{FH}$ population, which is likely enriched for T$_{H2}$ cells. Furthermore, the expressions of all the markers that we studied using immunofluorescence in IL-4–expressing T$_{FH}$ cells from IgG4-RD tissues (BCL6, ICOS, IL-4, CXCR5, BATF, GATA3, and PD1) were consistent with the RNA sequence observations on the tonsillar IL-4–secreting T$_{FH}$ cells.

### CD4$^+$CXCR5$^+$IL-4$^+$ T$_{FH}$ cells are mainly outside GCs in IgG4-RD and sometimes physically associate with AID-expressing B cells

As shown in Fig 4A, CD4$^+$CXCR5$^+$IL-4$^+$ T$_{FH}$ cells were abundant in affected IgG4-RD tissues and were mainly outside GCs but were also located in the light zone within GCs. Using parallel tissue sections, we noted that IgG4-positive B cells were abundant outside GCs in the same region in which IL-4–secreting T$_{FH}$ cells were observed. We quantitated IL-4–expressing T$_{FH}$ cells and IgG4-expressing B cells within and outside GCs and observed that the majority of both cell types reside outside GCs (Fig 4A and B).

It is generally accepted that CD40L-CD40 signaling induces AID and that specific cytokines target selected switch regions. Do the IL-4–expressing T$_{FH}$ cells and AID-expressing B cells make physical contact with one another? We examined lymph nodes and TLOs from IgG4-RD patients to determine whether ICOS$^+$ T$_{FH}$ cells are in physical contact with AID-expressing B cells or IgG4-expressing B cells in situ. AID-expressing B cells could be visualized outside GCs in IgG4-RD sections (Fig 4B and C); we noted that IgG4-expressing B cells outside GCs often express AID (Fig 4B). We also noted the existence of IL-4$^+$ICOS$^+$ T$_{FH}$ cells in cell–cell contact with AID-expressing B cells (Fig 4D). As has been reported earlier, much of the AID staining seen is cytosolic (Cattoretti et al, 2006).

---

**Figure 1. CD4$^+$ICOS$^+$IL-4$^+$ T$_{FH}$ cells are sparse in human SLOs.**
**(A)** Immunofluorescence staining of CD4 (red), ICOS (green), IL-4 (magenta), and DAPI (blue) in tissues from normal tonsils. The yellow broken line demarcates the area within GCs. The white broken line demarcates the area outside GC. **(B)** Immunofluorescence staining of CD4 (red), ICOS (green), IL-4 (magenta), and DAPI (blue) in tissues from normal tonsils. **(C)** Scatter plots depict the mean fluorescence intensity per cell quantified using TissueQuest software for each fluorescent antibody used to stain tissue from an IgG4-RD patient, normal tonsils, and normal mesenteric lymph nodes. **(D)** CD4$^+$ICOS$^+$IL-4$^+$ (red) and CD4$^+$ICOS$^+$IL-4$^-$ (green) cells were quantified in tissue from 12 tonsils. **(E)** Scatter plots depict the mean fluorescence intensity per cell quantified using TissueQuest software for each fluorescent antibody used to stain normal tonsils and mesenteric lymph nodes.

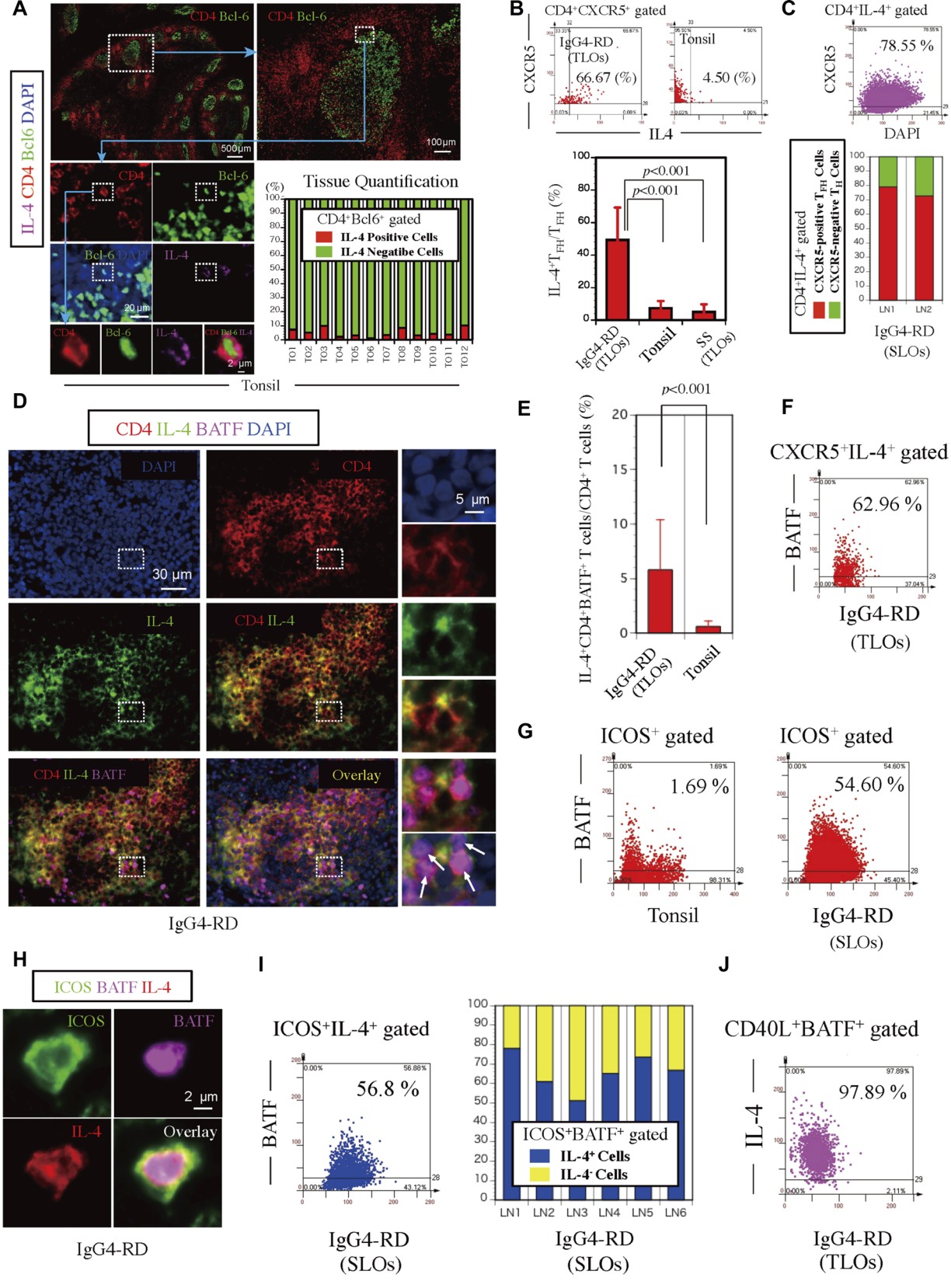

Furthermore, we noted the existence of IL-4$^+$ICOS$^+$ and BATF$^+$ICOS$^+$ T$_{FH}$ cells that are in cell–cell contact with IgG4-expressing B cells (Fig 4E and F), confirming visual contact with nuclear distance measurements. AID$^+$IgG4$^+$ B cells were also visualized within and outside GCs in TLOs in IgG4-RD.

### CD4$^+$CXCR5$^+$IL-4$^+$ T$_{FH}$ cells and CD4$^+$IL-4$^+$BATF$^+$ T cells are enriched in IgG4-RD, and their proportions are tightly linked to serum IgG4 levels and the proportion of IgG4-positive plasma cells in tissues

As shown in Fig 5A, only a small proportion of CD4$^+$CXCR5$^+$ T$_{FH}$ cells in tonsils, mesenteric and cervical lymph nodes from normal individuals, and also in or around TLOs from affected tissues from patients with SS synthesize IL-4. These data argue that IL-4 expression is not generally a part of the T$_{FH}$ phenotype in cells located in SLOs or TLOs. Instead, what is more likely is that a subset of activated T$_{FH}$ cells that express IL-4 expands considerably in a disease in which there is prominent IgG4 class switching, but these cells are not abundant around GCs in healthy people or in TLOs from other diseases in which there is no prominent class switching to the IgG4 or IgE isotypes. The proportion of T$_{FH}$ cells that express IL-4 in disease tissue correlates very strongly with the serum IgG4 levels in IgG4-RD patients but not with total serum IgG, IgE, or IgA levels (Fig 5B). A negative correlation with serum IgM levels was also observed (Fig 5B). The proportion of CD4$^+$CXCR5$^+$ cells that express IL-4 in IgG4-RD also correlates with the proportion of IgG4-expressing plasma cells in disease tissues (Fig 5C). The proportions of CD4$^+$BATF$^+$IL-4$^+$ T cells correlate with CD4$^+$CXCR5$^+$IL-4$^+$ T$_{FH}$ cells and with serum IgG4 levels (Fig 5D).

## Discussion

In T-dependent immune responses, it is widely accepted that specific cytokines induce the transcription of selected switch regions so that distinct Ig gene loci are targeted by AID for the induction of cytidine deamination and double-strand break formation. However, a link between subsets of T$_{FH}$ cells and specific isotype switching events has not been established.

We found that very few T$_{FH}$ cells in healthy tonsils or lymph nodes synthesize IL-4. These findings are broadly consistent with published data on human tonsillar T$_{FH}$ cells using different approaches (Ma et al, 2009; Bentebibel et al, 2011; Kroenke et al, 2012). Bentebibel et al (2011) reported that T$_{FH}$ cells from outside and within GCs secrete IL-4 when cocultured in vitro with B cells. However, as these data did not provide information at the single-cell level, the percentage of T$_{FH}$ cells capable of secreting IL-4 could not be surmised from this report. Kroenke et al used flow cytometry to quantitate intracellular cytokines in in vitro restimulated T$_{FH}$ cells from tonsils. Following restimulation with PMA and ionomycin, <3% of tonsillar T$_{FH}$ cells expressed IL-4 (Kroenke et al, 2012). Ma et al (2009) also used flow cytometry to measure cytokine expression by tonsillar T$_{FH}$ cells following PMA and ionomycin exposure and observed that up to 16% of CXCR5$^{hi}$ T$_{FH}$ cells expressed IL-4. Our in situ data, based on direct examination of IL-4 in T$_{FH}$ cells, rather than on in vitro restimulation, indicate that only a small fraction of T$_{FH}$ cells in normal human SLOs produce IL-4. These data suggest that IL-4–secreting T$_{FH}$ cells are relatively rare in human tonsils to begin with, and only a small fraction of healthy tonsillar T$_{FH}$ cells have presumably been restimulated in vivo to synthesize IL-4, as evidenced by our in situ studies. In IgG4-RD, in contrast, a very large proportion of T$_{FH}$ cells that can secrete IL-4 infiltrate tissues and SLOs, and because these cells have likely been restimulated in vivo, they can be seen to express IL-4 in our in situ analyses.

Human T$_{FH}$ subsets have only been described in the context of putative circulating memory CD4$^+$ T cells and not in a tissue context. Studies on T$_{FH}$ cells in disease have generally focused on an analysis of the blood. We examined human T$_{FH}$ cells in situ using multicolor quantitative immunofluorescence approaches, focusing on lymph nodes and TLOs in tissues of patients with a disease in which there is a pronounced switch to the IgG4 isotype. The proportions of CD4$^+$CXCR5$^+$IL-4$^+$ T cells, and their likely equivalent, CD4$^+$BATF$^+$IL-4$^+$ T cells, are markedly increased in IgG4-RD (with most IL-4–expressing T$_{FH}$ cells located in lymphoid cuffs just outside GCs), and these proportions correlate well with serum IgG4 levels and the proportion of IgG4-positive cells in disease tissues. Our data link IL-4–expressing T$_{FH}$ cells to IgG4 class switching and also indirectly argue that most class switching likely occurs outside GCs.

IL-4–secreting T$_{FH}$ cells have never been purified from lymphoid organs and cannot be distinguished in human lymphoid organs by the expression of specific chemokine receptors. To reliably characterize a specific cytokine-secreting T$_{FH}$ population, we used a specific IL-4 capture assay and compared the transcriptomes of IL-4–producing human tonsillar T$_{FH}$ cells with those of non-T$_{FH}$ cells from the tonsil that secrete IL-4 and human tonsillar PD-1$^{hi}$ T$_{FH}$ cells that do not secrete IL-4. IL-4–secreting T$_{FH}$ cells stand out as

---

**Figure 2. IL-4$^+$BATF$^+$ T$_{FH}$ cells are rare in normal SLOs but abundant in IgG4-RD.**
**(A)** Immunofluorescence staining of CD4 (red), Bcl-6 (green), IL-4 (magenta), and DAPI (blue) in tissues from normal tonsils. Quantification of CD4$^+$Bcl-6$^+$IL-4–positive T$_{FH}$ and CD4$^+$Bcl-6$^+$IL-4–negative T$_{FH}$ in 12 tonsils. **(B)** Scatter plots depict the mean fluorescence intensity per cell quantified using TissueQuest software for each fluorescent immunostain in IgG4-RD SMGs (G1) and normal tonsils. Quantification of CD4$^+$CXCR5$^+$IL-4$^+$ and CD4$^+$CXCR5$^+$ T$_{FH}$ cells in 17 IgG4-RD SMGs, 12 tonsils, and 7 SS salivary glands. The $P$-value is based on the Mann–Whitney $U$ test. **(C)** Almost all IL-4–expressing CD4$^+$ T cells in IgG4-RD lymph nodes were CXCR5$^+$ T$_{FH}$ cells. Scatter plots depict the mean fluorescence intensity per cell quantified using TissueQuest software for each fluorescent immunostain in IgG4-RD lymph node (LN1). The quantification of CD4$^+$IL-4$^+$ CXCR5–positive T$_{FH}$ and CD4$^+$IL-4$^+$CXCR5–negative non-T$_{FH}$ cells in lymph nodes from two patients with IgG4-RD (LN1 and LN2) is shown. **(D)** CD4$^+$IL-4$^+$BATF$^+$ T cells were enriched in IgG4-RD SMGs. Immunofluorescence staining of CD4 (red), IL-4 (green), BATF (magenta), and DAPI (blue) in IgG4-RD SMGs (G12). White arrows indicate CD4$^+$IL-4$^+$ BATF$^+$ T cells. **(E)** Quantification of CD4$^+$BATF$^+$IL-4$^+$ T cells and CD4$^+$ T cells in 17 IgG4-RD SMGs and 12 normal tonsils. The $P$-value is based on the Mann–Whitney $U$ test. **(F)** Scatter plots depict the mean fluorescence intensity per cell quantified using TissueQuest software for each fluorescent antibody used to stain normal tonsils and IgG4-RD lymph node (LN1). **(G)** Scatter plots depict the mean fluorescence intensity per cell quantified using TissueQuest software for each fluorescent immunostain in IgG4-RD SMGs (G3). **(H)** Immunofluorescence staining of ICOS (green), BATF (magenta), and IL-4 (Red) in IgG4-RD LNs (LN1). **(I)** Scatter plots depict the mean fluorescence intensity per cell quantified using TissueQuest software for each fluorescent immunostain in IgG4-RD LNs (LN1). The quantification of ICOS$^+$BATF$^+$IL-4–positive T and ICOS$^+$BATF$^+$IL-4–negative T cells in lymph nodes from six patients with IgG4-RD (LN1–LN6). **(J)** Scatter plots depict the mean fluorescence intensity per cell quantified using TissueQuest software for each fluorescent immunostain in IgG4-RD SMGs (G3).

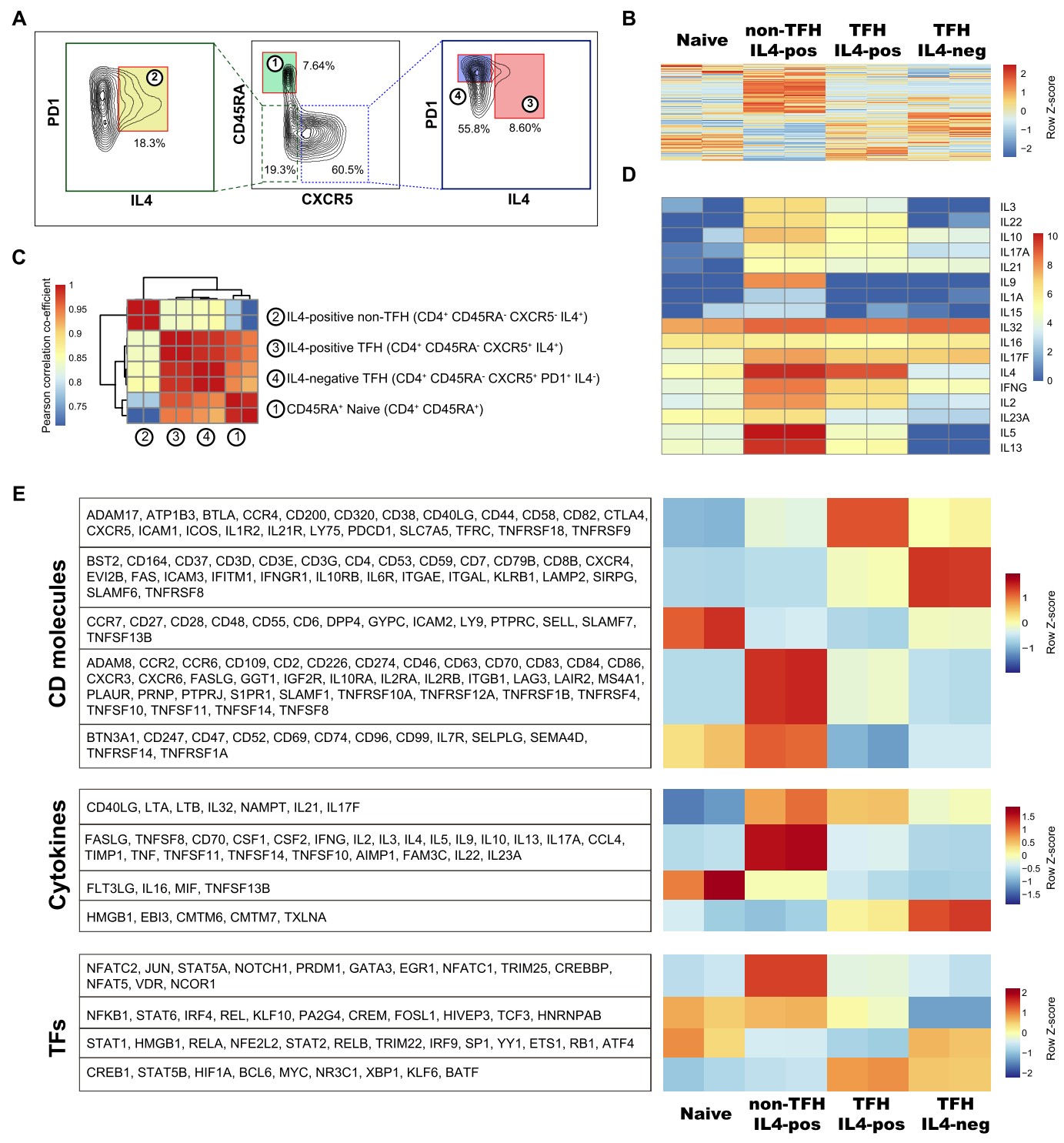

**Figure 3. Transcriptomic profiling of IL-4–producing T_FH cells from human tonsils.**
**(A)** Gating strategy used to sort IL-4–secreting and nonsecreting tonsillar CD45RA⁺CXCR5⁺ T_FH cells following anti-CD3/anti-CD28 stimulation. CD45RA⁺ CXCR5⁻ cells and CD45RA⁻CXCR5⁻IL-4⁺ cells were sorted as additional controls. **(B)** A heatmap of differentially expressed genes across all four conditions depicting the Z-scores for normalized expected read counts. **(C)** A correlation matrix of differentially expressed genes. **(D)** Expression pattern of individual ILs is depicted on a log scale. **(E)** Expression of CD molecules, cytokines, and transcription factors clustered into patterns using k-means. Z-scores of the expected read counts for each cluster are shown.

a distinct subset. It is reasonable to consider that these IL-4–secreting BATF-expressing T_FH cells, which do not express GATA-3, represent an activated T_FH population or subset in SLOs and TLOs that are expanded in disease that may contribute to class switching to IgG4. Unlike T_H2 cells, which produce IL-4, IL-5, and IL-13, the IL-4–producing tonsillar T_FH cells produce IL-4 and IL-10 but not IL-5

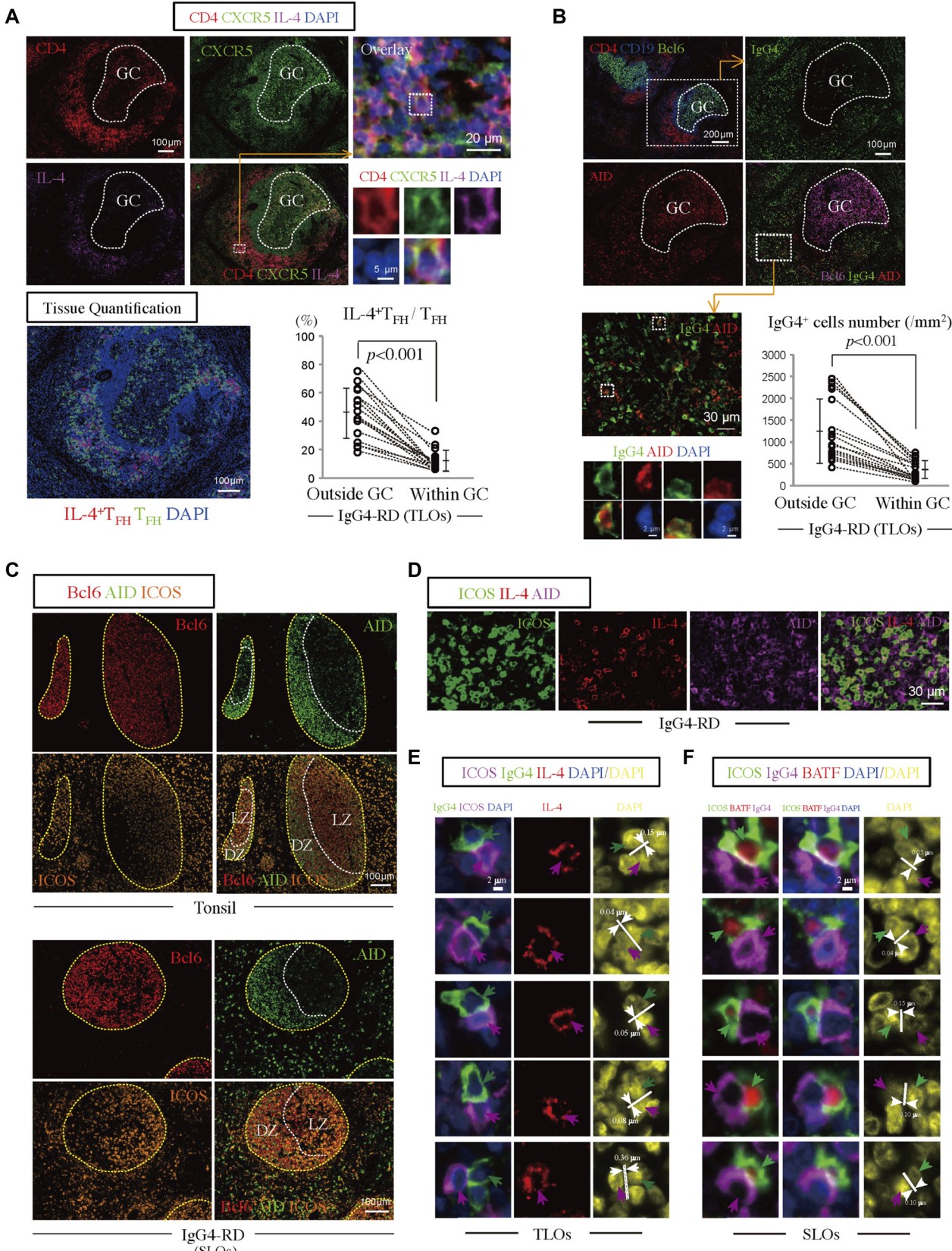

or IL-13. This subset of $T_{FH}$ cells has a unique transcriptional profile and a distinct set of surface markers that will facilitate future functional studies.

The mechanisms and signals by which antibodies class switch to IgG4 are poorly understood. CD4[+] T cells can activate IgM-positive B cells to switch to IgG4 and IgE in the presence of added IL-4 (Gascan et al, 1991). It has been argued based on in vitro studies that IL-10 may contribute to IgG4 class switching indirectly by somehow facilitating IL-4–mediated switching to IgG4 in preference to IgE (Jeannin et al, 1998). No molecular or cellular explanation has emerged for this in vitro phenomenon. Arguments have also been made that switching to IgG4 is a result of repeated division, and that switching progresses sequentially along chromosome 14 from IgG1 to IgG3 to IgG2 to IgG4 (Tangye et al, 2002; Jackson et al, 2014). We do not presume that IL-4–secreting $T_{FH}$ cells alone contribute to IgG4 class switching, although it is possible that the same subset of IL-4–secreting $T_{FH}$ cells that we have characterized might, in certain contexts, make enough IL-10 to facilitate this switching event. Perhaps more efficient purification of this $T_{FH}$ subset using surface markers inferred from our transcriptomic data will facilitate future functional studies on IL-4–secreting $T_{FH}$ cells. Tissue sources of IL-10 or of other factors that work in concert with IL-4 from these specific $T_{FH}$ cells will also need to be explored in future studies.

These studies are the first demonstration of an in situ expansion of what may be construed to be a human $T_{FH}$ subset. Many diseases exist in which IgE dominates in the absence of IgG4 and vice versa. Our data indicate that only a fraction of $T_{FH}$ cells in SLOs and TLOs, mainly outside GCs, express BATF and IL-4. However, we have shown here that these IL-4–expressing $T_{FH}$ cells accumulate in TLOs of patients with IgG4-RD. There is therefore a disease-specific increase in the accumulation of a subset of $T_{FH}$ cells and a tight association between the numbers of these cells and class switching, specifically to IgG4. Whether there are further subsets of BATF- and IL-4–expressing $T_{FH}$ cells that separately facilitate IgG4 or IgE class switching remains to be ascertained using single-cell approaches.

## Materials and Methods

### Study population

SMGs were obtained from 17 Japanese patients with IgG4-RD, from affected lymph nodes of 6 patients with IgG4-RD, and from lacrimal and salivary glands (LSGs) of 7 patients with active SS. In addition, 2 unaffected cervical lymph nodes, 3 mesenteric lymph nodes, and 12 healthy human tonsils were obtained, which were histologically normal. All these patients with IgG4-RD had been followed up between 2007 and 2015 at the Department of Oral and Maxillofacial Surgery of Kyushu University Hospital, a tertiary care center. Open SMG biopsies were obtained from patients with IgG4-RD (Moriyama et al, 2014). IgG4-RD was diagnosed according to the following criteria (Umehara et al, 2012): (i) persistent (longer than 3 mo) symmetrical swelling of more than two lacrimal and major salivary glands; (ii) high (>135 mg/dl) serum concentrations of IgG4; and (iii) infiltration of IgG4-positive plasma cells into tissue (IgG4[+] cells/IgG[+] cells >40%), as determined by immunostaining. All SMGs from patients with IgG4-RD had histopathologic features of IgG4-RD. Age, sex, serum Ig, and specific autoantibody levels of 17 patients with IgG4-RD (G1–G17) whose affected salivary gland biopsies were analyzed by ex vivo in situ immunofluorescence studies are summarized in Table S1A. Age, sex, serum Ig, and specific autoantibody levels of six patients with IgG4-RD (LN1–LN6) whose affected lymph node biopsies were analyzed by ex vivo in situ immunofluorescence studies are summarized in Table S2.

Each patient with SS exhibited objective evidence of salivary gland involvement based on the presence of subjective xerostomia and a decreased salivary flow rate, abnormal findings on parotid sialography, and focal lymphocytic infiltrates in the LSGs by histology (Vitali et al, 2002). All patients with SS were severe cases and had developed abundant TLOs in each salivary gland. None of the patients with IgG4-RD and SS had a history of treatment with steroids or other immunosuppressants, infection with HIV, hepatitis B virus, hepatitis C virus, or sarcoidosis, and none had evidence of malignant lymphoma at the time of the study. Age, sex, serum Ig, and specific autoantibody levels of patients with SS whose salivary gland biopsies were analyzed by in situ immunofluorescence are summarized in Table S1B.

Normal human mesenteric lymph node and tonsil patients were obtained from Massachusetts General Hospital. Normal human cervical lymph nodes were obtained from the Department of Oral and Maxillofacial Surgery of Kyushu University Hospital.

The study protocol was approved by the Ethics Committee of Kyushu University, Japan, and the Institutional Review Board at Massachusetts General Hospital. All patients provided written informed consent before participating in the study.

---

**Figure 4.   CD4[+]CXCR5[+]IL-4[+] $T_{FH}$ cells are abundant around GCs in IgG4-RD and sometimes physically contact AID-expressing B cells.**

**(A)** CD4[+]CXCR5[+]IL-4[+] T cells were enriched around TLOs in IgG4-RD SMGs. Immunofluorescence staining of CD4 (red), IL-4 (magenta), CXCR5 (green), and DAPI (blue) in IgG4-RD SMGs (patient G3). Quantification of CD4[+]CXCR5[+]IL-4[+] $T_{FH}$ cells and CD4[+]CXCR5[+] $T_{FH}$ cells comparing those outside and within GCs from each of five different areas in TLOs of 17 patients with IgG4-RD (G1–G17). The *P*-value is based on the Mann–Whitney *U* test. **(B)** IgG4[+]B cells were enriched outside GC, especially some these cells express AID. Immunofluorescence staining of CD4 (red), CD19 (blue), and Bcl6 (green) in IgG4-RD SMGs (patient G3). Immunofluorescence staining of AID (red), Bcl6 (magenta), IgG4 (green), and DAPI (blue) in IgG4-RD SMGs (patient G3). The numbers of IgG4[+] cells (per square micrometer) outside and within GCs were quantified from each of five different areas in TLOs of 17 patients with IgG4-RD (G1–G17). The *P*-value is based on the Mann–Whitney *U* test. **(C)** AID-expressing B cells outside GCs in IgG4-RD lymph nodes were abundant. Immunofluorescence staining of Bcl6 (red), AID (green), and ICOS (orange) in a normal tonsil and IgG4-RD lymph node. **(D)** Immunofluorescence staining of ICOS (green), IL-4 (red), and AID (magenta) in IgG4-RD SMGs. **(E)** Immunofluorescence staining of IgG4 (green), ICOS (magenta), IL-4 (red), and DAPI (blue) in TLOs with an IgG4-RD patient (G3). Magenta arrows indicate ICOS[+]IL-4[+] $T_{FH}$ cells. Green arrows indicate IgG4[+] B cells. A number of IL-4–expressing $T_{FH}$ cells and IgG4+ B cells formed close and extensive intercellular plasma membrane contacts. The distance was measured between the edge of each IgG4[+] B cell nucleus to the edge of the closest IL-4–expressing $T_{FH}$-cell nucleus, and an internuclear distance of less than 0.4 μm was indicative of a T-B conjugate. **(F)** Immunofluorescence staining of ICOS (green), IgG4 (magenta), BATF (red), and DAPI (blue) in the lymph nodes of an IgG4-RD patient. Green arrows indicate ICOS[+]BATF[+] $T_{FH}$ cells. Magenta arrows indicate IgG4[+] B cells. A number of BATF[+]ICOS[+] $T_{FH}$ cells and IgG4[+] B cells formed close and extensive intercellular plasma membrane contacts. The distance was measured between the edge of each IgG4[+] B-cell nucleus to the edge of the closest BATF[+]ICOS[+] $T_{FH}$-cell nucleus. All internuclear distances in this figure were <0.2 μm. LZ, light zone; DZ, dark zone.

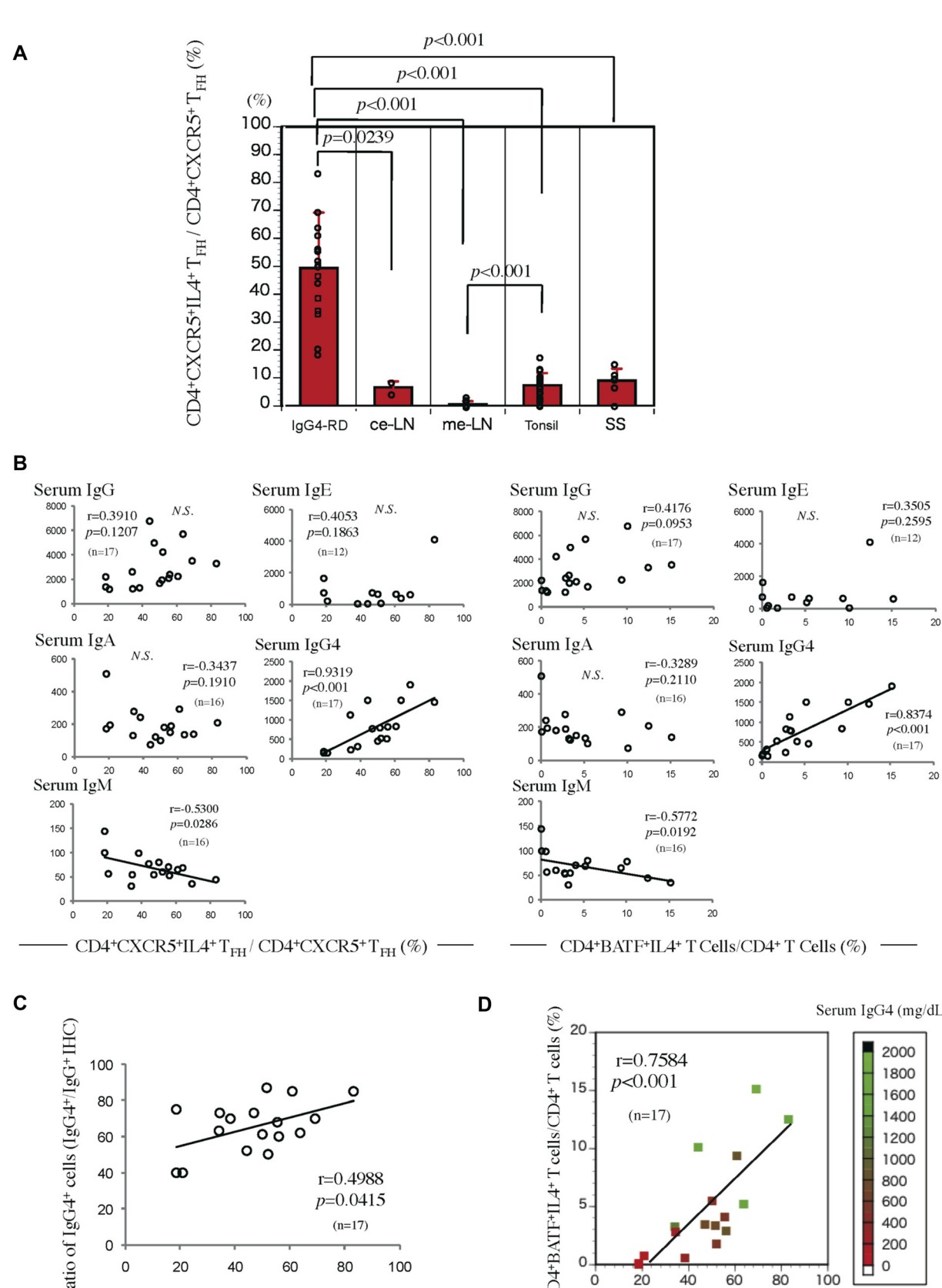

## Multicolor immunofluorescence staining

Tissue samples were fixed in formalin, embedded in paraffin, and sectioned. These specimens were incubated with the following antibodies: anti-AID (clone: ZA001; Invitrogen), anti-IgG4 (clone: ab109493; Abcam), anti-ICOS (clone: 89601; Cell Signaling Technology), anti-IL-4 (clone: MAB304; R&D Systems), GATA3 (clone: CM405A; Biocare), CXCR5 (clone: MAB190; R&D Systems), Bcl-6 (clone: CM410A, C; Biocare), BATF (clone: 10538; Cell Signaling Technology), CD4 (clone: CM153A; Biocare), and CD19 (clone: CM310 A,B; Biocare), followed by incubation with secondary antibodies using a SuperPicTure Polymer Detection Kit (Invitrogen) and an Opal 3-Plex Kit (Fluorescein, Cyanine3, and Cyanine5). The samples were mounted with ProLong Gold Antifade mountant containing DAPI (Invitrogen).

## Microscopy and quantitative image analysis

Images of the salivary gland specimens were acquired using the TissueFAXS platform (TissueGnostics). For quantitative analysis, the entire area of the tissue involved by the lymphoplasmacytic infiltrate was acquired as digital grayscale images in four channels with filter settings for FITC, Cy3, and Cy5 in addition to DAPI. Cells of a given phenotype were identified and quantitated using TissueQuest software (TissueGnostics), with cutoff values determined relative to the positive controls. This microscopy-based multicolor tissue cytometry software permits multicolor analysis of single cells within tissue sections similar to flow cytometry. The principle of the method and the algorithms used have been described in detail elsewhere (Ecker & Steiner, 2004).

## Evaluation of TLOs with IgG4-RD and SS

TLOs with GCs (Ruddle, 2014) were identified using multicolor immunofluorescence approaches (CD4, CD19, Bcl6, and DAPI). In this study, SMG and LSG tissue sections from 25 patients with IgG4-RD and 15 patients with severe SS were evaluated. Distinct Bcl-6$^+$ GCs were observed in affected IgG4-RD tissues. Bcl-6$^+$ GC B cells in TLOs were within B-cell follicles, delineated using antibodies to CD19. 17 of 25 patients (68%) with IgG4-RD had TLOs in affected salivary glands. Age, sex, serum Ig, and specific autoantibody levels of 17 patients with IgG4-RD (G1–G17) whose affected salivary gland biopsies were analyzed by in situ immunofluorescence are summarized in Table S1A. 7 of 15 patients (47%) with severe SS had TLOs in affected salivary glands. Age, sex, serum Ig, and specific autoantibody levels of patients with SS whose salivary gland biopsies were analyzed in this study are summarized in Table S1B.

The number of TLOs with GCs and the size of GCs in each TLO were evaluated in 4 mm$^2$ sections from five different areas of 17 patients with IgG4-RD (G1–G17) and 7 patients with SS (SS1–SS7) (Table S3).

## IL-4 capture assay and cell sorting

600 million cells from human tonsils were resuspended in Miltenyi magnetic-activated cell sorting buffer and stained with biotinylated anti-human CD19 (clone: HIB19; BioLegend) on ice for 25 min. Cells were washed once in magnetic-activated cell sorting buffer and incubated with anti-biotin microbeads (Miltenyi Biotec) for 25 min on ice. Cells were then loaded on two separate Miltenyi LS columns (at 300 million cells per column), and the flow-through was collected as the B cell–depleted fraction. These cells were spun down and stimulated overnight with plate-coated anti-CD3 (5 μg/ml) and anti-CD28 (5 μg/ml) antibodies. The stimulated cells were then enriched for IL-4–secreting cells using the vendor's protocol (IL-4 Secretion Assay Kit; Miltenyi Biotec, #130-054-101). The enriched IL-4$^+$ cell fraction was surface stained with antibodies against CD4 (BioLegend, #317420), CD45RA (clone: H100; BioLegend, #304122), CXCR5 (clone: J252D4; BioLegend, #356920), and PD1 (clone: EH12.2H7; BioLegend, #329924). The following populations were viably sorted using FACSAria2 (Becton Dickinson) directly into Qiagen RLT-plus buffer (with 1% 2-ME): (i) IL-4–secreting CD4$^+$CXCR5$^+$ T$_{FH}$ cells, (ii) IL-4–secreting CD4$^+$CD45RA-CXCR5$^-$ T$_{H2}$ cells, (iii) IL-4–negative CD4$^+$CXCR5$^+$PD1$^+$ T$_{FH}$ cells, and (iv) IL-4–negative CD45RA$^+$ cells.

## Trancriptomic analyses

Total RNA was isolated from the FACS-sorted cells using the RNeasy plus Micro Kit (Qiagen). RNA-sequencing libraries were prepared as previously described (Picelli et al, 2013). Briefly, whole transcriptome amplification and tagmentation-based library preparation were performed using the SMART-Seq2 protocol, followed by 35-bp paired-end sequencing on a NextSeq 500 instrument (Illumina). 5–10 million reads were obtained from each sample and aligned to the University of California, Santa Cruz hg38 transcriptome. Gene expression was calculated using RSEM as previously described (Li & Dewey, 2011). The EBSeq package was used to identify differentially expressed genes with a posterior probability of differential expression >0.95 (Leng et al, 2013). Our analysis focused on cytokines, transcription factors, and CD molecules. Cytokines and transcription factors were obtained using the gene ontology terms "GO:0003700" and "GO:0005125," respectively. Transcription factors pertaining to immune cells were filtered using a literature-based gene prioritization approach with the query terms "immune response," "immunity,"

---

**Figure 5. CD4$^+$CXCR5$^+$IL-4$^+$ T$_{FH}$ cells and CD4$^+$IL-4$^+$BATF$^+$ T cells are enriched in IgG4-RD, and their proportions are tightly linked to serum IgG4 levels and IgG4-positive plasma cells.**
**(A)** Quantification of CD4$^+$CXCR5$^+$IL-4$^+$ and CD4$^+$CXCR5$^+$ T$_{FH}$ cells in 17 IgG4- RD SMGs, seven SS LSGs, 12 tonsils, two cervical lymph nodes, and three mesenteric lymph nodes. The $P$-value is based on the Mann–Whitney $U$ test. **(B)** Correlations of the proportion of CD4$^+$CXCR5$^+$IL-4$^+$ T$_{FH}$ cells and CD4$^+$BATF$^+$IL-4$^+$ T cells in SMGs from patients with IgG4-RD and their serum IgG, IgA, IgE, IgM, or IgG4 levels. The $r$ and $P$-value were determined using Spearman's rank correlations. **(C)** Correlations of the proportions of CD4$^+$ CXCR5$^+$IL-4$^+$ T$_{FH}$ cells in SMGs from patients with IgG4-RD and their ratio of IgG4$^+$/IgG$^+$ cells (n = 17). The $r$ and $P$-value were determined using Spearman's rank correlations. **(D)** The frequency of CD4$^+$CXCR5$^+$IL-4$^+$ T$_{FH}$ cells in SMGs from patients with IgG4-RD (n = 17) correlated with the frequency of CD4$^+$BATF$^+$IL-4$^+$ T cells and serum IgG4 concentrations. The $r$ and $P$-value were determined using Spearman's rank correlations.

"T cells," and "lymphocytes" (Fontaine et al, 2011). The data discussed in this publication have been deposited in National Center for Biotechnology Information's Gene Expression Omnibus and are accessible through GEO Series accession number GSE111968.

## Statistical analyses

Differences between groups were determined using $\chi^2$ tests, $t$ tests, Mann–Whitney $U$ tests, and Spearman's rank correlations. All statistical analyses were performed using JMP Pro software, version 11 (SAS Institute) for Mac. $P$-values < 0.05 were considered statistically significant. Nonsignificant differences were not specified. In all figures, bar charts and error bars represent means ± SEM.

# Supplementary Information

# Acknowledgements

This work was supported by awards AI110495 (to S Pillai) and AI113163 (to VS Mahajan) from the National Institute of Health and supported by the Japanese Society for the Promotion of Science Postdoctoral Fellowships for Research Abroad to T Maehara and Mochida Memorial Medical and Pharmaceutical Foundation to T Maehara. The authors thank Eric Safai for help with the collection and processing of samples from Massachusetts General Hospital and Thomas Diefenbach of the Imaging Core at the Ragon Institute for help and advice.

## Author Contributions

T Maehara: conceptualization, formal analysis, investigation, and writing—original draft, review, and editing.
H Mattoo: conceptualization, investigation, and writing—original draft, review, and editing.
VS Mahajan: formal analysis, investigation, and writing—original draft, review, and editing.
SJH Murphy: investigation.
GJ Yuen: investigation and writing—review and editing.
N Ishiguro: resources.
M Ohta: resources.
M Moriyama: resources.
T Saeki: resources.
H Yamamoto: resources.
M Yamauchi: resources.
J Daccache: investigation.
T Kiyoshima: resources.
S Nakamura: resources.
JH Stone: resources.
S Pillai: conceptualization, formal analysis, supervision, funding acquisition, and writing—original draft, review, and editing.

## Conflict of Interest Statement

The authors declare that they have no conflict of interest.

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
