## [Reviewer comments · Life Science Alliance]

The expansion in lymphoid organs of IL-4+ BATF+ T follicular helper cells is linked to IgG4 class switching *in vivo*

Takashi Maehara, Hamid Mattoo, Vinay S. Mahajan, Samuel J. H. Murphy, Grace J. Yuen, Noriko Ishiguro, Miho Ohta, Masafumi Moriyama, Takako Saeki, Hidetaka Yamamoto, Masaki Yamauchi, Joe Daccache, Tamotsu Kiyoshima, Seiji Nakamura, John H. Stone and Shiv Pillai

DOI: 10.26508/lsa.201700050

Review timeline:	Submission Date:	8 March 2018
	1 st Editorial Decision:	9 March 2018
	Revision Received:	18 March 2018
	2 nd Editorial Decision:	19 March 2018

Report:

(Note: Letters and reports are not edited. The original formatting of letters and referee reports may not be reflected in this compilation.)

Please note that the manuscript was previously reviewed at another journal and the reports were taken into account in the decision-making process at Life Science Alliance. Since the original reviews are not subject to Life Science Alliance's transparent review process policy, the reports and author response cannot be published.

Please note that the manuscript was previously reviewed at another journal and the reports were taken into account in inviting a revision for publication at *Life Science Alliance* prior to submission to *Life Science Alliance*.

1st Editorial Decision

9 March 2018

Thank you for submitting your revised manuscript entitled "The expansion in lymphoid organs of IL-4+ BATF+ TFH cells is linked to IgG4 class switching *in vivo*" to Life Science Alliance. Your manuscript was reviewed at another journal before, and the referee reports of this previous round of review were confidentially transferred to us with your permission.

The reviewers who evaluated your work had noted that your study is robust, but does not provide functional data to examine the ability of IL-4-secreting TFH to mediate class switching to IgG4. After evaluating your manuscript, this is not a concern for publication in Life Science Alliance. Furthermore, we appreciate that although cytokine capture is a good method, it does not allow to capture enough IL-4 secreting TFH cells from tonsils in order to perform T-B collaboration assays.

We would therefore be happy to publish your paper in Life Science Alliance pending final revisions necessary to meet our formatting guidelines. I list below a few items you should pay attention to allow production of your manuscript. Congratulations on this very nice work!

Thank you for your attention to these final processing requirements.

2nd Editorial Decision

19 March 2018

Thank you for contributing your Research Article entitled "The expansion in lymphoid organs of IL-4⁺ BATF⁺ TFH cells is linked to IgG4 class switching in vivo". It is a pleasure to let you know that your manuscript is now accepted for publication in Life Science Alliance. Congratulations on this interesting work.